# Wear Resistance Design of Laser Cladding Ni-Based Self-Fluxing Alloy Coating Using Machine Learning

**DOI:** 10.3390/ma17225651

**Published:** 2024-11-19

**Authors:** Jiabo Fu, Quanling Yang, Oleg Devojno, Marharyta Kardapolava, Iryna Kasiakova, Chenchong Wang

**Affiliations:** 1State Key Laboratory of Rolling and Automation, Northeastern University, Shenyang 110819, China; 13029377910@163.com (Q.Y.); wangchenchong@ral.neu.edu.cn (C.W.); 2Faculty of Mechanical Engineering, Belarusian National Technical University, Khmelnitsky Str., 9, Build. 6, 220013 Minsk, Belarus; devoino-o@mail.ru (O.D.); margokardo@tut.by (M.K.); i.kosyakova88@gmail.com (I.K.)

**Keywords:** machine learning, alloy design, laser cladding, wear resistance, Ni self-fluxing alloys

## Abstract

To improve the collaborative design of laser cladding Ni-based self-fluxing alloy (SFA) wear-resistant coatings, machine learning methods were applied. A comprehensive database was constructed from the literature, linking alloy composition, processing parameters, testing conditions, and the wear properties of Ni-based SFA coatings. Feature correlation analysis using Pearson’s correlation coefficient and feature importance assessment via the random forest (RF) model highlighted the significant impact of C and B elements. The predictive performance of five classical machine learning algorithms was evaluated using metrics such as the squared correlation coefficient (*R²*) and mean absolute error (*MAE*). The RF model, which exhibited the best overall performance, was further combined with a genetic algorithm (GA) to optimize both composition and processing parameters collaboratively. This integrated RF-GA optimization system significantly enhanced efficiency and successfully designed multiple composition and process plans. The optimized alloy demonstrated superior wear resistance with an average friction coefficient of only 0.34, attributed to an enhanced solid solution strengthening effect (110 MPa) and increased hard phase content (52%), such as Ni₃Si, CrB, and NbC. These results provide valuable methodological insights and theoretical support for the preparation of laser cladding coatings and enable efficient process optimization for other laser processing applications.

## 1. Introduction

High-performance wear-resistant materials demonstrate substantial application potential and offer expansive prospects for development in the high-end equipment manufacturing industry [1]. However, optimizing multiple comprehensive properties of materials often complicates achieving optimal wear resistance. Applying high-performance coatings to the material surface can overcome inherent wear resistance limitations and ease the collaborative optimization of various material properties. Ni-based self-fluxing alloy (SFA) coatings have garnered widespread attention for their outstanding properties, such as excellent wear resistance, corrosion resistance, and oxidation resistance [2,3]. As the service environments for these materials become increasingly complex, conventional commercial Ni-based self-fluxing alloy coatings struggle to meet evolving demands. Therefore, developing advanced high-performance Ni-based self-fluxing alloy wear-resistant coatings is of critical importance [4].

Laser cladding technology has garnered significant attention in recent years for its ability to address common coating defects, such as porosity, unevenness, and weak adhesion, while producing denser coatings with robust substrate bonding. However, laser cladding technology, characterized by high-energy melting and rapid cooling to produce Ni-based self-fluxing alloy wear-resistant coatings, involves a complex control system with numerous process parameters [5,6]. The performance of laser cladding coatings is affected by the interplay of various parameters, such as laser power, scanning speed, spot size, and energy density. Additionally, there is a significant coupling between these parameters and the coating composition [7]. The optimal processing parameter combinations vary across different component systems, and understanding how each parameter and composition influences coating performance involves complex, multi-scale physical mechanisms [8]. Thus, precise optimization of the laser cladding process presents challenges both experimentally and in modeling. Experimentally, managing numerous key process parameters necessitates extensive orthogonal testing, while modeling faces issues such as complex models, numerous parameters, and high computational costs [9,10].

To address the challenges faced in traditional experiments and modeling methods, recent research has increasingly focused on artificial intelligence strategies based on data-driven approaches for precise quantitative design of complex component processes. A strong mathematical association between multi-dimensional input features and target outputs can be established by these AI strategies, bypassing complex and unclear physical mechanisms and enabling rapid and accurate prediction of the relationship between component process plans and final performance while reducing the need for extensive orthogonal experiments and avoiding cumbersome physical modeling [11,12,13,14,15]. In the field of Ni-based alloys, Conduit et al. [16] trained several artificial neural network (ANN) models on extensive datasets to predict mechanical properties such as fatigue life, yield strength, and fracture stress of nickel-based polycrystalline superalloys, subsequently using the models to design new alloys with excellent properties. Similarly, Suzuki et al. [17] utilized historical data on Ni-based SX superalloys from General Electric (GE) and applied various machine learning algorithms to develop predictive models for the alloys’ physical, mechanical, and environmental properties, facilitating efficient development and design. Additionally, Venkatesh and Rack [18] employed a back propagation neural network (BPNN) to predict the creep rupture life of Inconel 690 and Ni-based single crystal superalloys at 1000 °C and 1100 °C, achieving 100% and 90% prediction accuracy, respectively, within an acceptable error range of ±2 h. Yoo et al. [19] used Bayesian neural networks combined with the Markov chain Monte Carlo method to predict the creep life of Ni-based single crystal superalloys at high temperatures, achieving a prediction accuracy of 93.2% using material parameters such as chemical composition, testing temperature, and testing stress.

In the field of laser processing, research on process optimization based on artificial intelligence algorithms combined with high-throughput optimization algorithms has also been reported. For example, Barrionuevo et al. [20] developed a machine learning-assisted interpretable model that evaluated the impact of laser powder bed fusion processing parameters such as laser power, scanning speed, layer thickness, hatch distance, and material density on the hardness and wear property of printed metallic materials. Based on small sample data established experimentally, Zhen et al. [21] built a process optimization framework that couples a data-driven machine learning algorithm and a high-throughput multi-objective optimization algorithm to efficiently and reliably realize the collaborative optimization of taper and processing efficiency in femtosecond laser trepan drilling. The proposed framework avoids complex femtosecond laser processing parameters and internal mechanisms while avoiding time-consuming and labor-intensive repeated experiments and can be applied to other more complex laser material processing process collaborative optimization problems. However, the above-mentioned artificial intelligence research in the field of laser processing only focuses on the prediction and optimization of laser processing processes under a specific component system and does not involve the collaborative optimization of components and processes. Therefore, in the more complex problem of preparing wear-resistant coatings by laser cladding technology, how to reasonably build an artificial intelligence strategy to achieve collaborative design of components and processes is still a difficult issue worthy of further exploration.

To achieve efficient collaborative design for the composition and process of laser cladding Ni-based SFA wear-resistant coatings, machine learning methods were proposed and implemented in this study. A comprehensive database, established through literature research, included alloy composition, processing parameters, testing conditions, and friction coefficients. Pearson’s correlation coefficient (PCC) and random forest mean decrease in impurity (MDI) analysis revealed the influence of each feature on friction coefficient prediction, highlighting the importance of C and B elements. The random forest (RF) model demonstrated the best predictive performance among five classic machine learning algorithms, using metrics such as the squared correlation coefficient (*R²*) and mean absolute error (*MAE*). Integrated with a genetic algorithm (GA), RF was employed to optimize the composition and process, leading to superior wear-resistant coatings. The optimized alloy displayed superior wear resistance compared to existing coatings in the dataset at a computational level. These results provide valuable insights for optimizing laser-cladded Ni-based coatings and other laser processing applications.

## 2. Materials and Methods

### 2.1. Dataset and Data Preprocessing

The data used in this study were collected and extracted from published research studies [2,22,23,24,25,26,27,28,29,30,31,32,33,34,35], which were considered credible. These data contain a total of 54 samples on the wear resistance of laser cladding Ni-based self-fluxing alloy (SFA) coatings. A global overview of the dataset ranges is presented in Table 1. The dataset includes 17 input features, encompassing material characteristics, processing parameters, and environmental factors. Material characteristics cover 9 elemental properties (e.g., Cr, Fe, C, B, Si, Mo, Nb, Mn, and Cu) and 6 processing parameters (e.g., laser power, scanning speed, overlap ratio, spot diameter, number of cladded layers, and powder feeding speed). Environmental factors include testing load and sliding speed, both critical to the wear testing process. The friction coefficient serves as the model’s output to assess the wear resistance of the laser cladding Ni-based SFA coatings.

Given that the data features involve varying attributes, units, and orders of magnitude, all features were normalized using standard data preprocessing methods. In this study, the traditional standardization method, as shown in Equation (1), was employed to eliminate dimensional differences between features, thereby improving the training efficiency and prediction accuracy of the machine learning models. The correlations and importance of the data features were further assessed using correlation analysis methods, such as Pearson’s correlation coefficient (PCC), and feature importance was evaluated using the random forest mean decrease in impurity (MDI) method.
(1)$$z=\frac{x-\mu }{\sigma }$$

### 2.2. Construction of ML Framework

In this study, five classic machine learning algorithms were employed to develop predictive models for the hierarchical relationship of ‘component-process-wear resistance’ in laser cladding Ni-based SFAs, as shown in Figure 1. Based on the wear resistance dataset of laser cladding Ni-based SFAs detailed in Section 2.1, the selected algorithms include random forest (RF), multilayer perceptron (MLP), support vector regression (SVR), gradient boosting regression (GBR), and extreme gradient boosting (XGB). The dataset was split into training and test sets at an 8:2 ratio, and 50 random divisions were performed to mitigate the risk of prediction artifacts due to random partitioning and to reduce model uncertainty. During the modeling process, hyperparameters were optimized using the grid search method. The performance of each model was evaluated using the squared correlation coefficient (*R^2^*) and the mean absolute error (*MAE*), as defined by Equations (2) and (3). The model with the best performance metrics was selected for the subsequent reverse design of the composition and process.
(2)$$R^{2}=\frac{{\left(n\sum_{i=1}^{n}f\left(x_{i}\right)y_{i}-\sum_{i=1}^{n}f\left(x_{i}\right)\sum_{i=1}^{n}f\left(y_{i}\right)\right)}^{2}}{\left({n\sum_{i=1}^{n}f\left(x_{i}\right)}^{2}-{\left(\sum_{i=1}^{n}f\left(x_{i}\right)\right)}^{2}\right)\left({n\sum_{i=1}^{n}f\left(y_{i}\right)}^{2}-{\left(\sum_{i=1}^{n}f\left(y_{i}\right)\right)}^{2}\right)}$$
(3)$$MAE=\frac{1}{n}\sum_{i=1}^{n}\left|f\left(x_{i}\right)-y_{i}\right|$$

### 2.3. High-Throughput Coupled Design of Composition and Process

In this study, the optimal machine learning model was integrated with a genetic algorithm to iteratively optimize the composition and process of Ni-based SFA coatings. Before proceeding with the reverse design of the composition and process, it is essential to evaluate the impact of genetic algorithm parameters on the optimization process. Key parameters include initial population size, the number of generations before termination, crossover probability, and mutation probability. By fine-tuning these parameters, a more efficient optimization design system can be developed. Finally, using the optimal genetic algorithm parameters, the composition and laser cladding process of the Ni-based SFA coatings are randomly generated. The friction coefficient is predicted using the wear resistance predictive models under fixed test conditions, and the composition and process are iteratively optimized to achieve the reverse design.

## 3. Results

### 3.1. Feature Analysis

Correlation analysis of the multi-level data features of laser cladding Ni-based SFA coatings was performed using PCC. The PCC values range from −1 to 1, where an absolute value close to 1 indicates a strong linear correlation between two features, while a value near 0 signifies a weak linear correlation. The results, illustrated in Figure 2a, show that the absolute value of PCC between any two features is less than 1, suggesting that all features can be considered independent variables. Additionally, the absolute value of PCC for each feature relative to the output feature, the friction coefficient, is greater than 0, indicating that each feature contributes to the prediction of the friction coefficient. Moreover, it is evident that elements C, Nb, Mn, and B have higher PCC rankings, indicating their significant roles in optimizing the wear resistance of Ni-based SFA coatings. Specifically, the addition of element C promotes the formation of carbides such as Cr_7_C_3_, Cr_23_C_6_, and NbC, which enhances the hardness of the SFA coatings [27,36]. Nb contributes to increased solid solution strengthening, further improving wear resistance. Mn refines alloy grains and impedes dislocation movement, thus enhancing mechanical properties such as strength and hardness. Element B facilitates the formation of borosilicate Ni_3_Si and borides like Cr_2_B and CrB, which improve the wear resistance and stability of SFA coatings [25,28].

Feature importance was further assessed using MDI values from the random forest model, as shown in Figure 2b. The MDI rankings corroborate the PCC findings, with C and B elements being highly influential. Additionally, scanning speed, laser power, and overlap ratio also ranked prominently, as they affect grain size and dendrite growth during the laser cladding process, significantly impacting coating wear resistance [37,38].

### 3.2. Prediction of Friction Coefficient for Different ML Models

Before establishing a stable friction coefficient prediction model, the normalized data were shuffled and randomly divided, with 80% allocated to the training set and 20% to the test set. This process was repeated for a total of 50 random divisions. The performance of different machine learning models was then compared and analyzed based on the average *R^2^* and average absolute error (*MAE*) values.

Figure 3 displays the average *R^2^* and *MAE* values for different machine learning models across 50 random divisions. The results indicate that, except for the GBR model, other machine learning models achieved good predictions of the friction coefficient for laser cladding Ni-based self-fluxing alloys, with *R^2^* values exceeding 60% on the testing set and low *MAE* values (approximately 0.06). Notably, while the GBR model exhibited higher *R^2^* values on both the training and testing sets, it also had a higher *MAE*, suggesting a higher correlation with the true values but also greater prediction errors. This indicates that the GBR model may struggle with the complexity of predicting the friction coefficient from 17 input characteristics.

Additionally, the RF model showed a significant difference between its *R^2^* values on the training set (88.2%) and the testing set (62.4%), highlighting an overfitting issue and sensitivity to dataset division in small samples. In contrast, the MLP, XGB, and SVR models performed well on both training and testing sets, demonstrating strong generalization capabilities. Among these, the MLP model exhibited the best performance, with *R^2^* and *MAE* values of 63.7% and 0.064, respectively, and the smallest error bar, indicating the highest stability in predictions.

Since the dataset was randomly divided 50 times, most samples were included in both the training and testing sets multiple times, allowing each sample to yield multiple predicted values. Figure 4 illustrates the average prediction results of the friction coefficient and the optimal prediction results for the RF and MLP models across different datasets.

In the training set, both the RF and MLP models show that nearly all data points align closely with the dotted line representing a slope of 1, and have small error bars (see Figure 4a,d). This indicates excellent performance of both models within their respective training sets. However, in the testing set, the performance of the RF and MLP models was less stable compared to the training set. Some data points in the testing set deviated significantly from the dotted line, indicating larger errors. This suggests that both the RF and MLP models exhibit some sensitivity to dataset partitioning.

Notably, as shown in Figure 4c,f, the optimal RF and MLP models, tested under multiple random partitions, demonstrated high prediction accuracy on both the training and testing sets. Specifically, the RF model achieved an *R^2^* of 92.4% and an *MAE* of 0.035 on the testing set, outperforming the MLP model. This is further supported by the noticeable deviation of a few data points from the dotted line in Figure 4f, highlighting the superior suitability of the RF model for complex problems compared to the MLP model.

### 3.3. Coupled Design of Composition and Process

Based on the model selection and genetic algorithm parameter analysis discussed in Section 4.1 and Section 4.2, integrating the RF model proves to be a rational and effective approach for optimizing the friction coefficient and designing the process. Within the scope of the original dataset, the RF model demonstrates superior predictive performance across multiple random divisions, with *R^2^* values exceeding 80% for both the training and testing sets and an *R^2^* difference of less than 10% to mitigate overfitting (a total of 10 model sets). This indicates robust generalization capability and prediction accuracy, validating the selection of models with higher R² and minimal overfitting for subsequent genetic algorithm optimization.

Figure 5 illustrates the design outcomes using the optimal RF model with a fixed load of 30 N and a sliding speed of 10 mm/s [24,26,33]. The plot shows design results based on the optimal RF model (represented by the dots) compared to the original minimum friction coefficients from the database (depicted by the dotted lines). The results indicate that only one model yielded a friction coefficient close to the optimal value within the dataset, while the remaining nine process plans produced friction coefficients superior to those of existing alloys in the database. This outcome further substantiates the effectiveness and potential of RF models in optimizing design and application.

### 3.4. Validation of Newly Designed Coatings at Computational Level

Given the rapid cooling rate associated with laser cladding technology, the resulting Ni-based SFA coatings often exhibit a highly dense and uniform microstructure. Additionally, hard phases such as Ni_3_Si, boride, and carbide exhibit substantial strengthening effects in the Ni-based self-fluxing alloys [29,39]. To evaluate the strength of Ni-based SFA coatings, we calculated the solid solution strengthening effects of various designed alloys and their equilibrium volume fraction of the hard phase, as shown in Figure 6. The solid solution strengthening effects in the FCC matrix are calculated according to the following equations [40]:(4)$$\sigma_{\gamma }=\sum_{i}\left(\beta_{i}\times \sqrt{x_{i}^{\gamma }}\right)$$
where $\beta_{i}$ is a strengthening constant for the solution hardening of solute *i* in *γ* matrix [41] and $x_{i}^{\gamma }$ is the mole fraction of solute *i* in *γ* matrix, which are calculated using Thermo-Calc software (2023a) and the TCNI12 database.

The results reveal that Ni-based SFA coatings with different compositions generally exhibit a solid solution strengthening level of approximately 70 MPa and a hard phase content around 50%. Notably, the alloy with the best wear resistance in the dataset shows a superior combination of solid solution strengthening and hard phase content. Specifically, Alloy D2 from the designed alloys demonstrates a higher combination of solid solution strengthening (around 110 MPa) and hard phase content (approximately 52%) compared to Optimal Alloy 1 from the dataset. The composition and processing parameters of novel Alloy D2 designed by RF models and existing alloy Optimal 1 in the dataset are given in Table 2. This comparison underscores the significant advantages of using machine learning-based collaborative design processes in optimizing alloy performance.

## 4. Discussion

### 4.1. Comparison of Fitting Ability for Different ML Models

Ensuring a good fit between models and data is crucial, especially with less data in machine learning. To prevent overfitting or underfitting, a thorough evaluation of model fitting is essential. Underfitting occurs when a model shows high prediction errors and poor generalization in both training and testing sets. Figure 7 provides statistics on underfitting across different machine learning models based on 50 random divisions.

The results indicate that the RF, XGB, and MLP models have minimal underfitting, while the SVR model shows significant underfitting, exceeding 15 instances. Despite the general effectiveness of SVR on small datasets, it struggles with complex, high-dimensional, and interrelated data, such as laser cladding Ni-based SFA coatings, leading to notable underfitting. Conversely, the RF model demonstrates almost no underfitting, attributed to its robust ability to handle high-dimensional data and adaptability to complex datasets.

In addition to analyzing underfitting, assessing overfitting is crucial to ensure the robustness of machine learning models. Overfitting occurs when a model excels on the training set but performs poorly on the testing set, indicating poor generalization. In this study, we evaluated the degree of overfitting by examining the RF, XGB, and MLP models—those with less underfitting, as shown in Figure 7. The evaluation was based on the absolute difference between training and test set *R^2^* values. The results, presented in Figure 8, categorize the models into different overfitting intervals (0–10%, 10%–20%, >20%) and the distributions of R² on testing sets (60%–70%, 70%–75%, 75%–80%, >80%).

Figure 8a shows that the distribution of models across different overfitting intervals is quite similar for RF, XGB, and MLP, with no significant differences. However, Figure 8b highlights that the RF model stands out in the distribution of *R^2^* on the testing set. The RF model achieved an *R^2^* greater than 80% in approximately 12 cases, significantly outperforming the other models. The number of RF models with an *R^2^* below 75% was also lower, indicating that the RF model generally offered better prediction accuracy and generalization. This superior performance of the RF model can be attributed to its ensemble learning approach. RF uses multiple decision trees trained on different subsets of the data, allowing it to effectively capture nonlinear features and complex relationships. This approach enhances both prediction accuracy and generalization capability, making RF more robust compared to XGB and MLP models in this study.

### 4.2. Sensitivity Analysis of Genetic Algorithm Parameters

The parameterization of genetic algorithms is critical to achieving effective optimization in design tasks. Key parameters of genetic algorithms include population size, termination criteria, crossover probability, and mutation probability. Adjusting these parameters is essential for developing a more efficient optimization design system. Typically, the population size is chosen within the range of 20 to 200, and its selection profoundly affects the algorithm’s performance. A larger population size can enhance exploration of the design space, although it increases computational demands, while a smaller initial population size may result in convergence to a local optimum. In this study, we initially performed population initialization using genetic algorithms, incorporating nine elemental characteristics, six processing parameters, and two environmental factors. Subsequently, crossover and mutation operations were applied to generate offspring, and selection strategies were used to form a new population. This iterative process was repeated, integrating the RF model to achieve the optimal design.

Specifically, we set the crossover probability at 0.7 and the mutation probability at 0.01 and limited the maximum number of generations to 500. We conducted three genetic algorithm evaluations with varying initial population sizes (20, 40, 60, 80, and 100) and analyzed their effects on convergence generation (EGN), convergence time (TS), and global search capability (GSC). EGN indicates the number of iterations required for the algorithm to reach the optimal solution, reflecting the number of iterations needed to identify the optimal friction coefficient. TS represents the duration required for the algorithm to stabilize and find the optimal friction coefficient. GSC assesses the algorithm’s ability to identify optimal solutions throughout the search space, specifically indicating the number of optimal solutions discovered within the population.

The results are summarized in Table 3. It is observed that as the population size increases, both EGN and TS also increase. This is attributed to the larger number of individuals that require evaluation, crossover, and mutation, which enhances computational complexity and broadens the search space. The GSC peaks at 62.33 with a population size of 40, at which point EGN is minimized at 284.33. When the population size is increased to 100, convergence generation reaches a maximum of 349.33, and global search capability significantly decreases to 52.67. Overall, a population size of 40 provides a high global search capability (62.33) and a reasonable convergence time (3.67 s).

When configuring the crossover probability (P_c_) and mutation probability (P_m_) in genetic algorithms, it is common to select a larger crossover probability, typically ranging from 0.5 to 1.0, while keeping the mutation probability within a smaller range of 0.001 to 0.05. A higher crossover probability can accelerate the generation of offspring and facilitate the emergence of high-quality solutions. However, excessively high crossover probabilities may disrupt the inheritance of traits. Conversely, a mutation probability that is too low can result in slow convergence, whereas a mutation probability that is too high may compromise the algorithm’s randomness and effectiveness. Hence, finding the optimal combination of crossover and mutation rates through experimentation is crucial.

Figure 9 illustrates the optimization convergence curves of the genetic algorithm under varying combinations of crossover and mutation probabilities, with other parameters held constant. The figure demonstrates that adjustments to these probabilities can significantly impact the algorithm’s convergence performance. A low crossover probability often leads to slower convergence and poor global search performance, while a high crossover probability may result in premature convergence to sub-optimal solutions. Therefore, achieving optimal results necessitates selecting a suitable balance between crossover and mutation rates to enhance both the exploratory and convergent capabilities of the algorithm.

As shown in Figure 9a, when the crossover probability is set to 0.5, the algorithm typically requires more than 150 generations to converge to a satisfactory solution across the four mutation probabilities tested. Notably, with mutation rates of 0.01 and 0.07, convergence to the optimal solution takes up to 400 generations. Low mutation rates (e.g., 0.01) lead to a sluggish introduction of genetic diversity, resulting in inefficient searches, while high mutation rates (e.g., 0.07) may disrupt well-established solutions, leading to overly random searches and ineffective convergence. Similarly, Figure 9b,c show that with crossover probabilities of 0.6 and 0.7, the algorithm generally requires around 300 generations to identify high-quality solutions with friction coefficients below 0.35. At mutation rates of 0.03, 0.05, and 0.07, the desired low friction coefficient solutions were not found until approximately 350 generations, while at a mutation rate of 0.01, the solution was only achieved towards the end of the generation cycle. When the crossover probability is increased to 0.8, as depicted in Figure 9d, the algorithm rapidly produces solutions, often reaching sub-optimal solutions in fewer than 50 generations, indicating high update speed and search efficiency. Particularly, with a mutation rate of 0.03, the optimal solution with a friction coefficient below 0.35 can be attained within approximately 50 generations. Consequently, a combination of P_c_ = 0.8 and P_m_ = 0.03 effectively balances efficiency and accuracy, ensuring the genetic algorithm performs optimally in identifying the best parameter set.

### 4.3. Comparison Between the Newly Designed Alloy and the Original Alloy

A comparative analysis of the composition of the designed Alloy D2 versus various Ni-based SFA coatings from the dataset was performed using the simplified distance function, as defined by Equation (5) [42].
(5)$$D=\left(1-\sqrt{\frac{\sum_{j=1}^{N}{\left(Y_{d,j}-Y_{o,j}\right)}^{2}}{N}}\right)\times 100\%$$
where *N* is the number of elements in the alloy system and $Y_{d,j}$ and $Y_{o,j}$ represent the concentration of the $j_{th}$ element in the designed alloy and the original alloy in dataset, respectively. The results of compositional correlation between the designed Alloy D2 and Ni-based SFA coatings used in the dataset are calculated and shown in Figure 10. Alloy D2 shows no direct correlation with any of the 18 original alloys, with correlation coefficients ranging from 65% to 90%. Among these, the highest correlation (91.15%) is with original alloy 14, while the lowest (69.98%) is with Optimal Alloy 1, highlighting the novelty of the designed alloy.

Figure 11a illustrates the composition comparison between the newly designed Alloy D2 and the best original alloy. Compared to the original best Alloy 1, Alloy D2 exhibits notable increases in alloying elements, including significant rises in the content of C and Si, as well as the introduction of Mo, Nb, and Cu, which contribute to enhanced wear resistance [39]. Specifically, Alloy D2 features a C content increased from 0.03 wt.% to 0.71 wt.% and incorporates 3.4 wt.% Nb. This addition is reflected in the phase composition by an increase in NbC and the emergence of a graphite phase, as shown in Figure 11b, which improve the hardness and self-lubricating properties of the Ni-based SFA coating [32]. Additionally, the Si content rises from 1.5 wt.% to 4.21 wt.%, and the phase composition shows an increase in Ni₃Si in the ordered phase. The introduction of Fe, Mo, and Cu further enhances the solid solution strengthening effect of the alloy [22].

## 5. Conclusions

In summary, to achieve efficient collaborative design of the composition and process for laser cladding Ni-based SFA wear-resistant coatings, machine learning methods were employed and applied in this study.

(1)A comprehensive database was created through literature research, encompassing alloy composition, process parameters, and wear resistance of coatings for laser cladding Ni-based self-fluxing alloys. Feature correlation analysis using PCC and feature importance assessment based on MDI from a random forest model revealed that each feature in the dataset significantly impacts the prediction of the friction coefficient, with C and B elements being particularly influential.(2)Subsequently, the prediction performance of five classic machine learning algorithms was compared based on the average *R²*, the value of *MAE*, and the ability of generalization. The RF model, demonstrating the best overall performance, was selected and integrated with a genetic algorithm for the collaborative optimization of the composition and process of the Ni-based SFA coatings.(3)The combination of the RF model and genetic algorithm facilitated the development of a highly efficient optimization design system. This system successfully generated various composition and process plans for Ni-based SFA coatings. The final designed Alloy D2 demonstrates an average friction coefficient of only 0.34 under test conditions with a fixed load of 30 N and a sliding speed of 10 mm/s. Its excellent wear resistance can be attributed to a high content of strengthening phases, such as Ni₃Si, CrB, and NbC, of approximately 52%, along with a significant solid solution strengthening effect, estimated at around 110 MPa.(4)The experimental validation of the newly designed Alloy D2 is expected to be conducted in future work to verify the rationality of the composition and process, along with an in-depth analysis of its wear resistance mechanism. These findings offer valuable methodological insights and theoretical support for the preparation of laser cladding coatings, facilitating efficient process optimization in broader laser processing applications.

## Figures and Tables

**Figure 1 materials-17-05651-f001:**
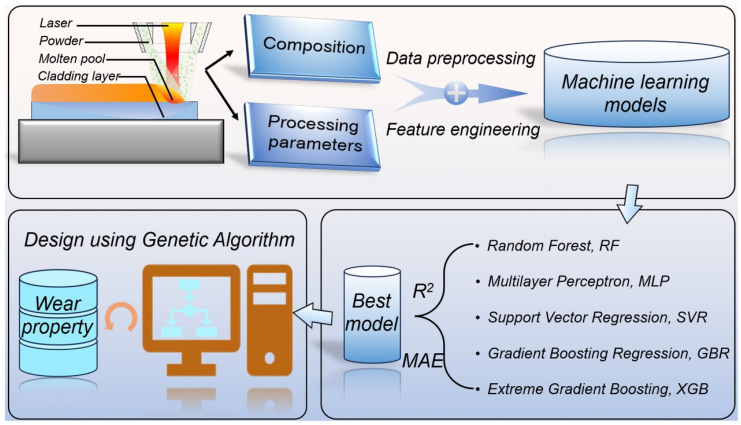
Framework for the design of laser cladding Ni-based self-fluxing alloys utilizing machine learning methods.

**Figure 2 materials-17-05651-f002:**
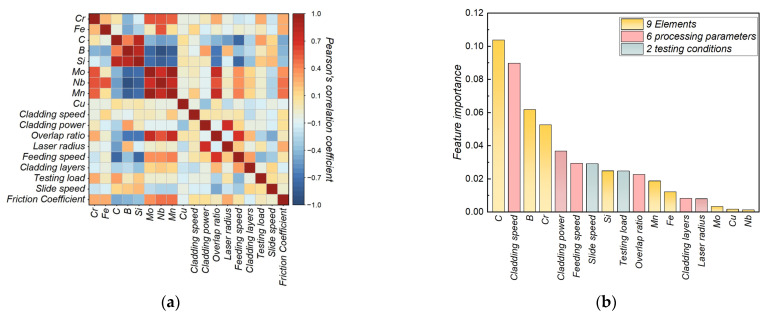
Results of feature analysis. (**a**) PCC between all features; (**b**) feature importance of input features.

**Figure 3 materials-17-05651-f003:**
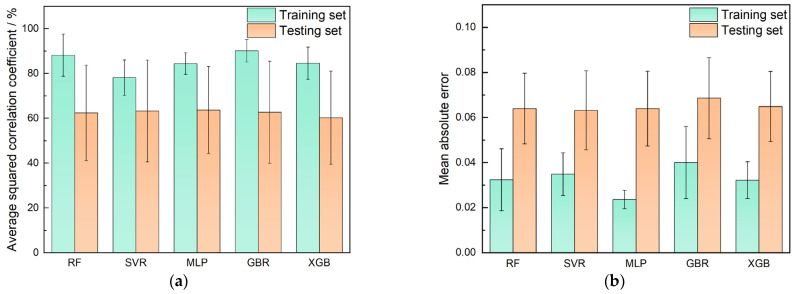
Performance of different ML models. (**a**) Mean *R^2^* and (**b**) *MAE*.

**Figure 4 materials-17-05651-f004:**
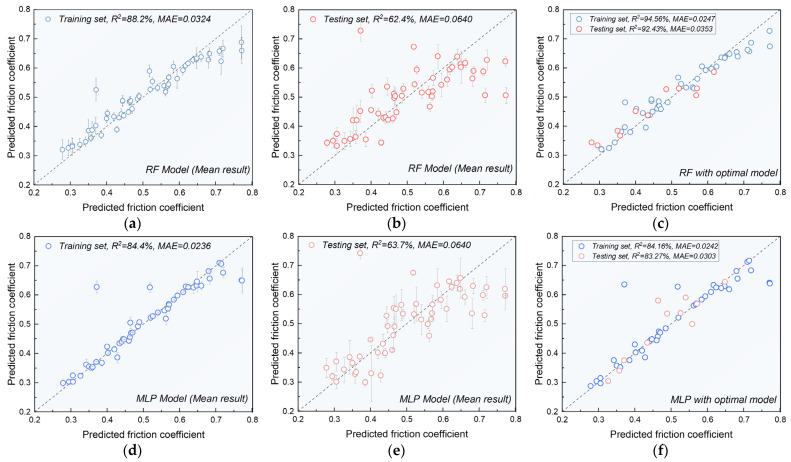
Experimental values vs. predicted values. Mean results for RF model in (**a**) training set and (**b**) testing set, and (**c**) optimal result for RF model; mean results for MLP model in (**d**) training set and (**e**) testing set, and (**f**) optimal result for MLP model.

**Figure 5 materials-17-05651-f005:**
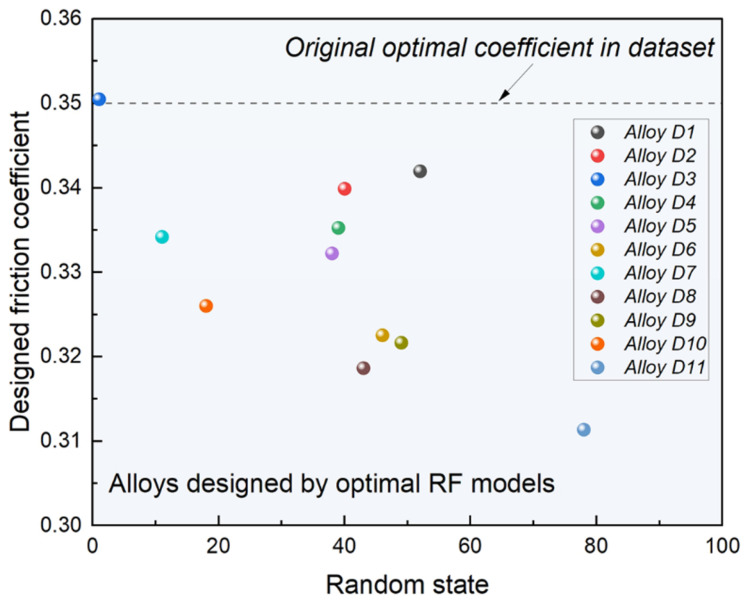
Optimization and design results using RF model and GA.

**Figure 6 materials-17-05651-f006:**
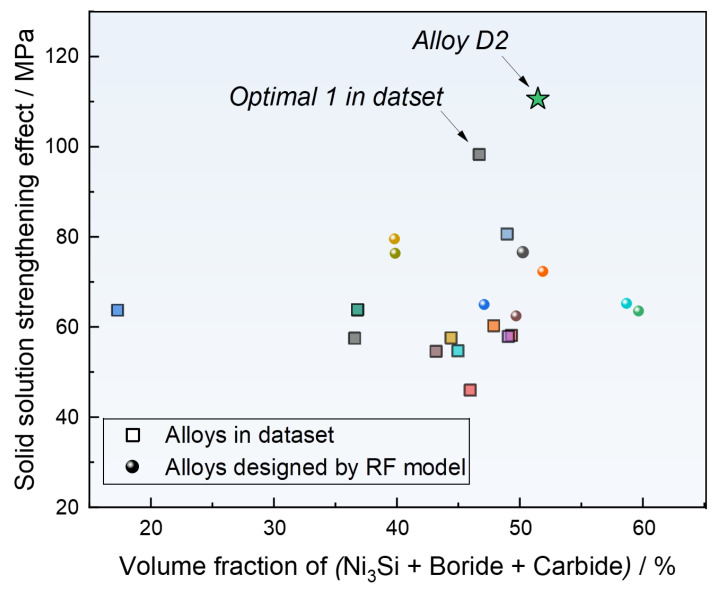
Comparison of hard phases and undesirable phases at lower temperatures between designed alloys (as indicated by spheres with different colors) and optimal alloys in dataset (as indicated by blocks with different colors) using Thermo-Calc software (2023a) and TCNI12 database.

**Figure 7 materials-17-05651-f007:**
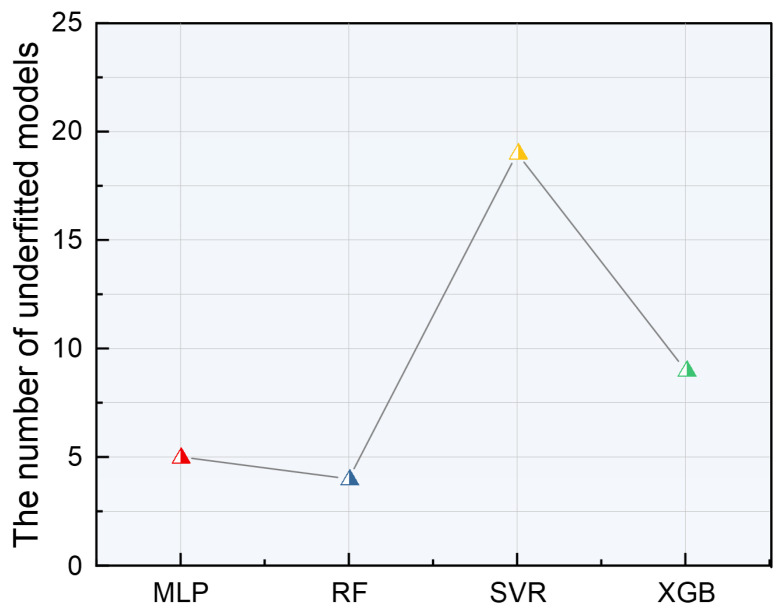
Number of underfitted models under multiple divisions of different machine learning models.

**Figure 8 materials-17-05651-f008:**
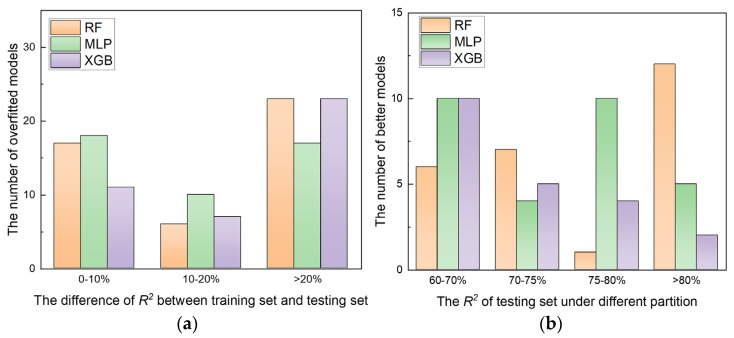
Number of overfitted models (**a**) and better models (**b**) under multiple divisions of different machine learning models.

**Figure 9 materials-17-05651-f009:**
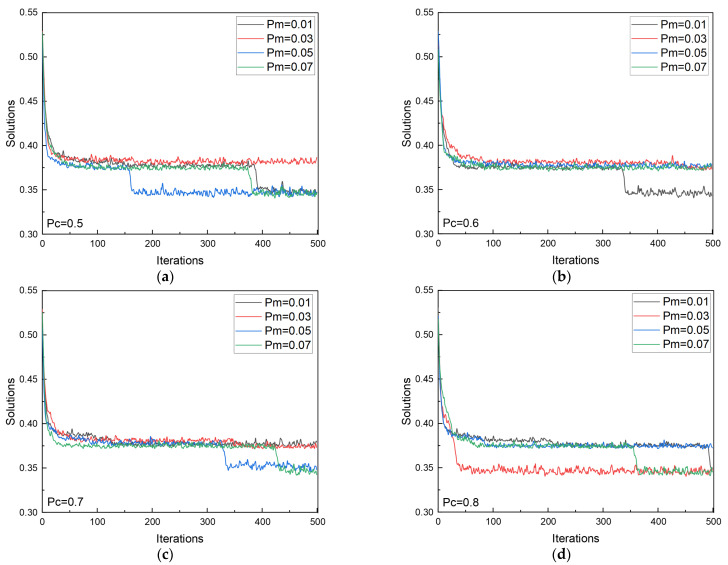
The change of solution in the calculation process of GA. (**a**) P_c_ = 0.5; (**b**) P_c_ = 0.6; (**c**) P_c_ = 0.7; (**d**) P_c_ = 0.8.

**Figure 10 materials-17-05651-f010:**
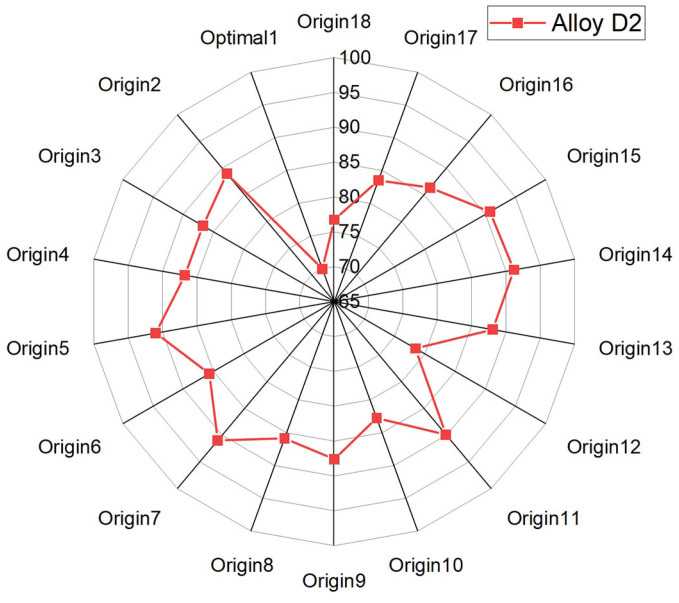
Comparison of compositional correlation between designed Alloy D2 and Ni self-fluxing alloys used in dataset using distance function.

**Figure 11 materials-17-05651-f011:**
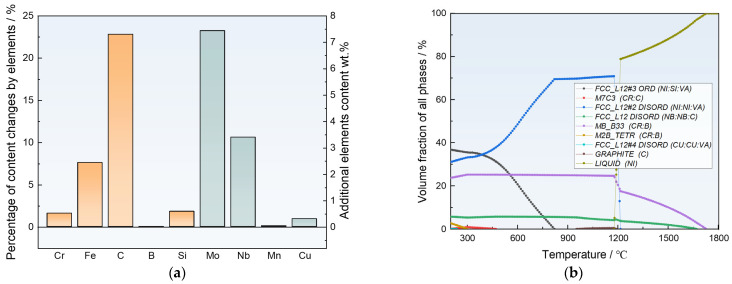
(**a**) Elemental content changes in Alloy D2 compared to the Optimal Alloy 1 in the original dataset and (**b**) phase configurations of Alloy D2.

**Table 1 materials-17-05651-t001:** Ranges of input and output parameters in the complete dataset.

	Features	Maximum	Minimum	Mean	Standard Deviation
Inputs	Cr (wt.%)	21.87	6	15.00	4.29
	Fe (wt.%)	20.596	0	5.06	5.58
	C (wt.%)	1.21	0	0.44	0.36
	B (wt.%)	4	0	2.42	1.25
	Si (wt.%)	5.83	0	3.18	1.77
	Mo (wt.%)	9.5	0	1.27	2.81
	Nb (wt.%)	5	0	0.82	1.72
	Mn (wt.%)	0.39	0	0.06	0.12
	Cu (wt.%)	15	0	0.53	2.16
	Scanning speed (mm/min)	76860	150	4301	14,167.01
	Laser power (W)	3000	800	1912	464.79
	Overlap ratio (%)	85	28	43.89	15.42
	Laser radius (mm)	6	0.85	2.82	1.22
	Powder feeding speed (g/min)	32.3	0.96	15.14	7.88
	Cladding layers	3	1	1.74	0.80
	Testing load (N)	100	5	29.72	25.26
	Sliding speed (mm/s)	2620	5	400.11	628.47
Output	Friction coefficient	0.772	0.2782	0.51	0.13

**Table 2 materials-17-05651-t002:** Composition and processing parameters of novel Alloy D2 designed by RF models and existing alloy Optimal 1 in dataset.

Features	Alloy D2	Optimal 1 in Dataset
Ni	62.1	89.09
Cr	15.47	6
Fe	3.26	0.38
C	0.71	0.03
B	3.07	3
Si	4.21	1.5
Mo	7.44	0
Nb	3.4	0
Mn	0.03	0
Cu	0.3	0
Scanning speed (mm/min)	250	300
Laser power (W)	2206	2000
Overlap ratio (%)	50	50
Laser radius (mm)	2.3	4
Powder feeding speed (g/min)	19.4	30
Cladding layers	2	3
Friction coefficient	0.34	0.35

**Table 3 materials-17-05651-t003:** The calculated results under different population sizes.

Initial Population Size	20	40	60	80	100
EGN	304	284.33	316	344.33	349.33
TS	3.28	3.67	3.80	4.20	4.54
GSC	59.33	62.33	54.33	59.33	52.67

## Data Availability

The raw data supporting the conclusions of this article will be made available by the authors on request.

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
