# Peer review of "Wear Resistance Design of Laser Cladding Ni-Based Self-Fluxing Alloy Coating Using Machine Learning"

_materials, 2024, doi:10.3390/ma17225651_

Round 1
Reviewer 1 Report
Comments and Suggestions for Authors
The manuscript's authors “Wear resistance design of laser cladding Ni-based self-fluxing alloy coating using machine learning”.
The authors provided a sufficient state of art study in the introduction section with good balance of new and older studies. They made a sufficient overview of the last researchers in the field, highlighted key aspects and clearly stated the focus of their study. The authors presented a detailed description of research methods and a detailed explanation of the purpose of the implementation of different artificial neural network methods.
In results section authors presented a detailed discretion of gained results with sufficient level of analysis
Authors follow the Materials journal guidelines and form the paper structure properly several aspects should be improved:
1. (Chapter 4) In this chapter, the author tends to make the discussion of gained results. However, they didn't present an actual discussion with other studies. It is recommended to reconsider this chapter via adding the discussion of presented results with up to 5-7 sources.
2. Though the conclusion section corresponds to the aim of the research, it is recommended to add more specific results and outcomes and avoid general phrases like “The final designed Alloy D2 exhibited superior wear resistance compared to existing coatings in the dataset at computational level..” (Line 446-448). Measurable parameters of wear resistance coating are desired.
3. It is highly recommended to declare the direction of feature research in the conclusion section. So far, the experimental validation of calculated characteristics has not been presented in the current study.
Reviewer 2 Report
Comments and Suggestions for Authors
This paper studied “Wear resistance design of laser cladding Ni-based self-fluxing alloy coating using machine learning”.
1- In abstract section, the authors should add more results that they found. Make it concise form possibly with some numerical results.
2- Add more detail about the Laser cladding process of laser cladding Ni-based self-fluxing alloy coating. If possible, some images of this process should be used for the explanation.
3- Explain more details about Fig. 1. The dimensional and the image of workpiece (Ni-based self-fluxing alloy) should be added.
4- In section 2.1 Dataset and Data Preprocessing, the data used in this study were sourced entirely from literature surveys (ref. 2, 22-35). The author should get permission from the the copyright holder (Copyright Permissions), before you can use the data from the previous research.
5- Increase the size of letters in Fig. 2
6- It is too bulky. Make it concise form possibly with some numerical results.
Comments on the Quality of English LanguageThe English could be improved to more clearly express the research.
Reviewer 3 Report
Comments and Suggestions for Authors
A database has been compiled containing the necessary information for the design of Ni-based claddings and coating used for laser applications. The framework for the design of laser cladding Ni-based self-fluxing alloys utilizing machine learning methods is presented. The database contains from the composition of the allow (textural to the wear properties of the final products. Various alternative machine learning algorithms are employed for the design and their performance is assessed via metrics like the reflections and the mean square error. Random forest model is preferred which, after being further optimized with use of genetic algorithm, gives enhanced efficiency. A comparison of hard phases and undesirable phases at lower temperature between designed alloys and optimal alloys in dataset is provided while the elemental content changes in certain alloys compared to the best alloy in the original dataset together with phase configurations of the allow are presented.
The paper considers an interesting problem, it contains a variety of approaches and the performance of the final designs is better than the ones being available in the beginning. However, certain improving modifications are required to become publishable at MDPI Materials; in particular:
(A) A better novelty statement is required. On what front is the proposed technique better compared to the competing ones? A more extensive discussion is necessary on the performance of alternative approaches.
(B) An important topic that should be elaborated in the revised version is the effect of potential anisotropy on the microstructure and properties of various alloys. How do the supported phases as function of temperature as well as the reflectivity are influenced? How the findings of the present paper are related to the performance of anisotropic samples [1] or the way they interact with light [2]?
(C) The paper would get substantially strengthen if the authors actually constructing and measuring the optimal samples instead of just simulating them.
[1] Anisotropic porous designed polymer coatings for high-performance passive all-day radiative cooling, iScience, 2022.
[2] Study of an electrically anisotropic cylinder excited magnetically by a straight strip line, Progress In Electromagnetics Research, 2007.
Author Response
We thank the reviewer for the valuable comments. Additional clarification in response to Comment 2 is provided in the attachment.

Round 2
Reviewer 2 Report
Comments and Suggestions for Authors
The quality of paper has been improved.
It can be accepted for publication.
Comments on the Quality of English LanguageThe quality of English is fine.
Author Response
We thank the reviewer for the detailed comments, which have greatly contributed to improving the clarity and quality of our manuscript.